# Gliosarcoma Is Driven by Alterations in PI3K/Akt, RAS/MAPK Pathways and Characterized by Collagen Gene Expression Signature

**DOI:** 10.3390/cancers11030284

**Published:** 2019-02-27

**Authors:** Bartosz Wojtas, Bartlomiej Gielniewski, Kamil Wojnicki, Marta Maleszewska, Shamba S. Mondal, Pawel Nauman, Wieslawa Grajkowska, Rainer Glass, Ulrich Schüller, Christel Herold-Mende, Bozena Kaminska

**Affiliations:** 1Laboratory of Molecular Neurobiology, Nencki Institute of Experimental Biology, 02-093 Warsaw, Poland; b.wojtas@nencki.gov.pl (B.W.); b.gielniewski@nencki.gov.pl (B.G.); k.wojnicki@nencki.gov.pl (K.W.); m.maleszewska@nencki.gov.pl (M.M.); 2Laboratory of Bioinformatics, Nencki Institute of Experimental Biology, Warsaw 02-093, Poland; s.mondal@nencki.gov.pl; 3Department of Neurosurgery, Institute of Psychiatry and Neurology, Warsaw 02-957, Poland; pnauman69@gmail.com; 4Department of Pathology, The Children’s Memorial Health Institute, Warsaw 04-730, Poland; w.grajkowska@czd.pl; 5Neurosurgical Research, University Clinics, 80539 LMU Munich, Germany; rainer.glass@med.uni-muenchen.de; 6Institute of Neuropathology, University Medical Center, 20251 Hamburg-Eppendorf, Germany; u.schueller@uke.de; 7Research Institute Children’s Cancer Center Hamburg, 20251 Hamburg, Germany; 8Department of Pediatric Hematology and Oncology, University Medical Center Hamburg-Eppendorf, 20251 Hamburg, Germany; 9Division of Experimental Neurosurgery, Department of Neurosurgery, University of Heidelberg, 69120 Heidelberg, Germany; h.mende@med.uni-heidelberg.de

**Keywords:** gliosarcoma, glioblastoma, transcriptome, PI3K/Akt, RAS/MAPK, mutation, collagen

## Abstract

Gliosarcoma is a very rare brain tumor reported to be a variant of glioblastoma (GBM), IDH-wildtype. While differences in molecular and histological features between gliosarcoma and GBM were reported, detailed information on the genetic background of this tumor is lacking. We intend to fill in this knowledge gap by the complex analysis of somatic mutations, indels, copy number variations, translocations and gene expression patterns in gliosarcomas. Using next generation sequencing, we determined somatic mutations, copy number variations (CNVs) and translocations in 10 gliosarcomas. Six tumors have been further subjected to RNA sequencing analysis and gene expression patterns have been compared to those of GBMs. We demonstrate that gliosarcoma bears somatic alterations in gene coding for PI3K/Akt (*PTEN*, *PI3K*) and RAS/MAPK (*NF1*, *BRAF*) signaling pathways that are crucial for tumor growth. Interestingly, the frequency of *PTEN* alterations in gliosarcomas was much higher than in GBMs. Aberrations of PTEN were the most frequent and occurred in 70% of samples. We identified genes differentially expressed in gliosarcoma compared to GBM (including collagen signature) and confirmed a difference in the protein level by immunohistochemistry. We found several novel translocations (including translocations in the *RABGEF1* gene) creating potentially unfavorable combinations. Collected results on genetic alterations and transcriptomic profiles offer new insights into gliosarcoma pathobiology, highlight differences in gliosarcoma and GBM genetic backgrounds and point out to distinct molecular cues for targeted treatment.

## 1. Introduction

Gliosarcoma is a very rare, glioblastoma IDH-wildtype brain tumor that accounts for less than 0.5% of all intracranial tumors and is preponderant in patients aged between 40 and 60 years old, with a slight male predominance [1,2]. Gliosarcoma is considered to be a glioblastoma (GBM) IDH-wildtype subtype [3], however some recent studies suggest that it may be a different entity [2,4]. Clinically gliosarcoma progresses rapidly and patients have a 3% higher risk of mortality as compared to GBMs [5]. A retrospective study from the MD Anderson Cancer Center found that the median overall survival (OS) of primary gliosarcoma is 17.5 months [6]. A recent French multi-center study estimated the median OS of gliosarcoma as only 13 months and reported that temozolomide (TMZ) chemotherapy is not associated with an improvement in OS compared to radiation [7]. Gliosarcoma is characterized by a mixture of gliomatous and sarcomatous elements [8]. The gliomatous compartment shows features of GBM as it is anaplastic, frequently spatially separated, characterized by invasion of the dura, leptomeninges, and hyperplastic or hypertrophied blood vessels. The sarcomatous compartment shows signs of malignant transformation with nuclear atypia, mitotic activity and packs of spindle cells. Some gliosarcoma cases manifest mesenchymal differentiation with collagen deposition [9]. In clinical practice, gliosarcoma are generally managed similarly to GBM, however, several clinical features such as a propensity to extracranial metastasis, distinct radiological features and worse prognosis than GBM, suggest that gliosarcoma may be a distinct clinico-pathological entity [10].

Early attempts to characterize the molecular background of gliosarcoma tumors have been based on case reports. In recent years, genetic alterations in primary and secondary gliosarcoma have been characterized [6,11,12,13]. Molecular analysis revealed a high incidence of *TP53* mutations and, rarely, *EGFR* and *IDH* mutations [6,11,12,13]. Gliosarcomas are similar to primary GBM in their molecular profiles and exhibit a similar rate of *NF1*, *RB1* and *PTEN* alterations. However, *TP53* mutations are more frequent and the rate of *EGFR* amplification/overexpression is lower in gliosarcoma as compared to GBM [6,11,14]. Several unique copy number alterations were identified in gliosarcoma and a subset of alterations developed specifically in the sarcomatous component [13].

Although histopathologically distinct, gliomatous and sarcomatous compartments of gliosarcomas share specific genetic alterations and likely derive from a common clonal origin [13,15,16]. The analysis of the gliomatous and sarcomatous components of eight gliosarcomas by comparative genomic hybridization after microdissection revealed that both components shared 57% of the detected chromosomal imbalances [17]. However, the number of chromosomal alterations in gliosarcomas was significantly lower than that in GBMs, indicating a higher genomic stability in gliosarcomas [17].

Despite certain differences in molecular profile and histological characteristics compared to GBM, gliosarcoma is typically treated similarly to GBM. A number of early phase clinical trials are testing targeted therapies in unique molecularly characterized subsets of patients (baskets [18]). Availability of information regarding the molecular setting of an individual gliosarcoma may increase therapeutic opportunities for patients. Using target enrichment and next generation sequencing with a panel of 664 cancer-related genes, we determined somatic mutation/indel profiles and copy number variations (CNVs) in 10 gliosarcomas, and performed transcriptomic analyses of six gliosarcomas by RNA sequencing. Moreover, transcriptomic data were employed to find genetic translocations. The expression of selected proteins was studied by immunohistochemistry. The obtained profiles of genomic alterations and gene expression patterns were used for the comparative analyses of gliosarcoma and GBM. Our results provide new insights into the molecular pathobiology of gliosarcoma.

## 2. Results

### 2.1. A Spectrum of Somatic Mutations and Indels in Gliosarcomas

In search of somatic mutations, we analyzed DNA samples from 10 gliosarcoma specimens and matched blood DNA samples. We sequenced the target-enriched exomic regions of 664 cancer-related genes. We found at least one missense/nonsense, non-tolerated/not-benign somatic mutation with a minimal variant allele frequency (VAF) of 20% in 6 out of 10 gliosarcoma samples (Table 1). The well-known BRAF V600E mutation was found in the GSM1 sample with a lower penetration in the GSM10 sample (Table 1 and Figure 1). Two different, missense mutations in the PIK3CA gene were detected in the GSM1 sample. Two missense PTEN mutations were detected in GSM2 and GSM9 specimens. The *TP53* mutation was detected in two gliosarcoma specimens. The mismatch mutation in the mismatch repair gene *MSH6* was found in the GSM3. Some mutations have not been detected in the primary Varscan2 analysis, but were found after a secondary manual inspection. In summary, somatic mutations were found in 7/10 gliosarcoma samples. A fraction of detected somatic variants was validated and confirmed by Sanger sequencing. Selection of variants for Sanger sequencing was done primarily for *PTEN* and *NF1* variants as they were found to be the most frequent ones, depending on the availability of samples, some additional gene variants were validated.

Somatic coding indels in *NF1* (3/10 gliosarcoma samples), *PTEN* (2/10), *RB1* (2/10) and *PIK3R1* (2/10) genes with a minimal penetration of 15% (within a given sample) were found in 5/10 samples (Table 2 and Figure 1). One indel in the *NF1* gene in the GSM4 specimen was found with a lower penetration, but as it occurred in a very well covered region (~300×), it was recognized as a very confident indel call.

### 2.2. Chromosomal Gain/Loss in Gliosarcomas

All gliosarcoma tumors had unbalanced copy numbers when compared to their normal reference DNA (blood DNA samples). The segmentation analysis of called CNVs revealed five chromosomal gains or losses: chromosome seven gain (four samples), chromosome 10 loss (three samples), chromosome 12 loss (two samples), chromosome 11 gain (one sample), and chromosome 19 loss (one sample) as seen in Figure 2. Moreover, a possibility of focal deletions in the PTEN and *NF1* foci was evaluated, and the *PTEN* focal deletion was detected in two samples (Figure 3, Appendix A).

### 2.3. Summary of Somatic Alterations in Gliosarcoma

In summary, *PTEN* was the most frequently altered gene in gliosarcomas, as 7/10 specimens had somatic mutations, indels or focal deletions in the *PTEN* gene region. The *NF1* gene was altered in 3/10 gliosarcomas due to indels. Somatic mutations in the *NF1* gene have been reported in human GBM tumors [19,20], amongst which nonsense mutations, splice site mutations, missense changes, and frameshift indels were present. Several of these mutations have been reported as germline alterations in neurofibromatosis patients, thus are likely inactivating [21,22]. Though, *EGFR* mutations/indels were not detected in gliosarcomas, the chromosome seven amplification (the region where the *EGFR* gene is located) was very frequent and occurred in 4/10 gliosarcomas, however no focal amplification was observed in the *EGFR* locus. Somatic alterations in the *RB1* were found in 3/10 gliosarcomas, while mutations in the *TP53* gene were detected only in 2/10 samples. Activating missense mutations of the *PIK3CA* gene, occurring in regions coding for the adaptor binding domain (ABD) as well as the C2 helical and kinase domains, have been reported in GBM [19]. We found frequent alterations in *PIK3* genes in 4/10 gliosarcoma specimens, with the PIK3CA gene being mutated twice in one sample and the *PIK3R1* gene harboring the indel in two samples. Novel missense somatic mutations (based on the COSMIC database) have been found in genes: *CDKN2C*, *HDAC11*, *PAX6*, *PTCH1*, *USP6*, and *WSC1*, novel somatic stop codon mutations have been found in *NKX2-1* and *OPTN* genes. Novel frame shift indels were detected in *NF1* and *PIK3R1* genes in the patient GSM3, both with a very high penetration above 50%, suggesting that these two mutations are driving neoplastic transformation. A summary of genetic alterations in gliosarcomas is presented in Figure 3, while references of existing variations from current databases (COSMIC, dbSNP) are included in Appendix A. All somatic alterations in *NF1* and *PTEN* genes were validated by Sanger sequencing; a majority, except one, were positively validated, * details are depicted in Appendix A. Location of *PTEN* and *NF1* genetic alterations in the context of respective protein structures is depicted in Figure 3B,C.

We estimated tumor purity using absCNSeq library in R. We found that the purity of the tested samples varied from 0.35–0.72 (details in the Appendix A, which is in the accepted range. Thus, in our analysis, we do not expect frequent false negative results due to tumor sample dilution by normal tissue components.

### 2.4. Transcriptome Analysis of Gliosarcoma and Comparison to GBM

Transcriptomic profiles may provide a link between cellular phenotypes and their molecular underpinnings. We performed the transcriptomic analysis of gliosarcomas by RNA-sequencing (RNAseq), and compared their gene expression profiles to those of GBMs to estimate the similarity between those two entities. Six gliosarcoma samples were subjected to RNAseq analysis and the results were compared to data collected from eight GBM IDH-wildtype specimens. The analysis of TCGA GBM subtypes revealed that most gliosarcoma samples fall into a mesenchymal category (4 out of 6), while one sample falls into a classical subtype category, and one sample (GSM1) is between classical and mesenchymal categories (Appendix A).

The Volcano plot (Figure 4A) shows a striking lack of similarity of the gene expression patterns between gliosarcoma and GBM samples. Using a t-test, we identified 1303 significantly differentially expressed genes (false discovery rate (FDR) < 0.05) between gliosarcoma and GBM, including 210 up-regulated genes and 1093 down-regulated genes in gliosarcomas compared to GBM. As depicted by the Volcano plot in Figure 4A, gliosarcomas showed the higher expression of some genes, most likely related to the sarcomatous tissue. Several mRNAs: *MLYCD*, *CFAP54*, *ATP9B*, and *TMEM212* were significantly downregulated in gliosarcomas when compared to GBMs (Figure 4B). More interestingly, we found several genes highly up-regulated in gliosarcoma, including the genes encoding proteins putatively related to differentiation: *UNC5B*, a metallopeptidase (*ADAMTS4*) and a collagen *COL18A1*, in comparison to GBMs (log2 fold change; Figure 4C). *UNC5B* is a receptor for Netrin-1, an axon guidance factor. Netrin-1-UNC5B signaling is implicated in the regulation of invasion and angiogenesis in medulloblastoma [23]. When compared to GBMs, gliosarcomas show up-regulation of some genes, ranging up to a 6-fold difference on a log2-scale (Figure 4C) and most top differentiating genes encode collagens (*COL1A1*, *COL6A1*, *COL6A2*, *COL6A3*, *COL1A2*, *COL3A1*) (Figure 4C).

Based on collected RNAseq data, we performed functional (pathways, gene groups) analysis of genes differentially expressed between GBMs and gliosarcomas. Our analysis revealed that many genes coding for proteins involved in the regulation of gene transcription have lower expression in gliosarcomas than in GBMs (Figure 5A). Moreover, the focal adhesion functional group (Table 3) was differently regulated in gliosarcomas and GBMs. It could be attributed to the different collagen signature in GBMs and gliosarcomas (Figure 4C). All functional groups of genes that significantly discriminated GBMs and gliosarcomas are depicted in Table 3.

### 2.5. Translocations in Gliosarcomas and Their Biological Consequences

Genomic rearrangements that give rise to oncogenic gene fusions may indicate actionable targets for cancer therapy. We present a systematic analysis of gene fusions among gliosarcomas. Gene fusions were identified by RNA-sequencing and TopHat Fusion software. We considered gene fusions to be pathogenically relevant when producing divergent gene expression, or as functionally important events. Overall, translocations were discovered in 5/6 samples processed for RNA-seq. One translocation within chromosome seven between *RABGEF1* and *GTF2IRD1P1* genes was present in three samples (Table 4 and Figure 5). Interchromosomal translocation between chromosome eight (ENSG00000254349) and chromosome 12 (*PCBP2*) was detected (Table 4 and Figure 5). RAB guanine nucleotide exchange factor 1 (*RABGEF1*) is implicated in the development of certain human cancers [24,25]. *PCBP2*, Poly(C) binding protein 2, plays an important role in post transcriptional and translational regulation of various signaling molecules through direct binding to single stranded poly(C) motifs. *PCBP2* has been reported to play a critical role in the development of multiple human tumors [26,27]. Several transcripts were annotated to less known mRNA sequences described in ensembl database, without any functional importance assigned up to date. *RABGEF1*–*GTF2IRD1P1* fusion occurring in three gliosarcoma samples was confirmed by nested PCR analysis and Sanger sequencing (Appendix A).

### 2.6. Collagens Expression in Gliomatous and Sacromatous Compartments

Distinction of two components of gliosarcoma requires combined histochemical and immunohistochemical staining. The gliomatous component of gliosarcoma (Figure 6, the left panel) marked by GFAP (glial fibrillary associated protein) staining displayed foci of necrosis surrounded by dense tumor cells; this palisading necrosis is a characteristic feature of GBM. Negative Gomori and positive GFAP staining (Figure 6) defined the glial neoplastic component. Only walls of blood vessels were Gomori positive (Figure 6). Collagen 6A3 (COL6A3) and collagen 3A1 (COL3A1) were abundantly expressed in blood vessels, without visible staining in other fields. The sarcomatous component of gliosarcoma (Figure 6, right panel), containing mainly densely packed, long bundles of mesenchymal neoplastic cells, showed a strong Gomori staining. Spindle cells were a characteristic element of gliosarcoma. Collagen 6A3 positive areas were abundant in this compartment and positive staining was not restricted to blood vessels. COL3A1 staining was confined to blood vessels (Figure 6) showing differences from COL3A1 staining in the gliomatous area. COL3A1 is most likely a very specific marker of gliosarcoma compartment as it was found not detected in any of tissues in immunochemistry results from Appendix A.

## 3. Discussion

Gliosarcoma is considered to be an IDH-wildtype subtype of GBM and gliosarcoma patients are typically treated according to the standards established for GBM. Previous genomic studies of gliosarcoma have focused on a limited panel of genes known to be mutated in GBM [6,14], recently gliosarcoma tumors were also profiled with whole exome sequencing [11]. In the present study, we characterize 10 gliosarcomas by the target enrichment sequencing of 664 cancer-related genes and transcriptomic analyses. Here, we demonstrate the comprehensive landscape of cancer related alterations in gliosarcoma, a rare, enigmatic and highly aggressive brain tumor. Most gliosarcoma tumors had somatic alterations of PIK3/Akt (*PTEN*, *PI3K*) and RAS/MAPK (*NF1*) signaling pathways that are crucial for tumor growth [28,29]. *PTEN* somatic mutations/indels were detected in 50% (5/10) of specimens and this frequency is higher than previously reported: 14% (2/14) and 38% (8/21), respectively [6,14]. In fact, together with focal deletions, the frequency of somatic alterations in *PTEN* was 70% (7/10), suggesting that the *PTEN* alteration is crucial for gliosarcoma development (Figure 3). PTEN (a phosphatase and tensin homolog deleted on chromosome-10) is a negative regulator of mitogenic signaling mediated by class 1 phosphatidylinositol 3-OH kinases (PI3K). Deletions or mutations in the *PTEN* gene are frequent events and are associated with therapeutic resistance in GBM [19]. *PTEN* in mesenchymal GBMs is deleted/mutated in round 50% of cases according to TCGA [19]. In our cohort the *PTEN* gene is more frequently altered (70%), however a small sample size limits the strength of our findings and a larger validation cohort would be desired. If in the future, a therapy targeting *PTEN* gene will be introduced into clinics, gliosarcoma would be the perfect tumor for a clinical trial.

We found frequent alterations in *PIK3* genes in 3/10 gliosarcoma specimens, with the *PIK3CA* gene being mutated twice in one gliosarcoma and the *PIK3R1* gene harboring indel-events in two samples. The *PIK3CA* gene encodes a catalytically active PI3K p110α protein, and the *PIK3R1* gene encodes a regulatory p85α protein forming the PI3 kinase complex. Mutations of the *PIK3CA* gene have been reported in human GBMs [19,30] and could represent an alternative event to *PTEN* mutations for deregulating this key, glioma-relevant pathway.

The frequency of *NF1* somatic indels in this study was 30% (3/10), and no single nucleotide mutation was found. The frequency of *NF1* mutations in gliosarcoma was previously described to be 18% (2/11) [6]. The *NF1* gene is frequently deleted in a mesenchymal type of glioblastoma, but the overall frequency of the *NF1* gene deletion/mutation was reported to be around 30% in mesenchymal GBMs [19], similar to gliosaroma samples. Interestingly, the frequency of *PTEN* or *NF1* alterations was much higher in gliosarcoma than in GBMs, reported to be 41% for *PTEN* and only 10% for *NF1* in GBM patients [19]. We found the frequency of somatic alterations in the genes such as *NF1*, *PTEN*, *TP53*, and *PI3K* similar as in other studies [6,11,14], however in our cohort, *PTEN* somatic alterations are more frequent than previously reported and they are likely to drive gliosarcoma development.

We also found the BRAFV600E mutation present in two patients (2/10, 20%), which is in contrast to the reported lack of BRAFV600E immunoreactivity in 48 gliosarcoma tumor sections [31], but in agreement with a recent finding [32]. Activating BRAF-V600E mutations are recurrently found in pediatric glial and glioneuronal brain tumors [12,33] and specific inhibitors are entering clinical trials. Finding the BRAFV600E mutation opens a new possibility to treat gliosarcoma patients with targeted therapies against the mutated BRAF protein.

Somatic alterations in the *RB1* gene were found in 30% of samples (3/10). We found a lower frequency of *TP53* mutations (30%, 3/10) than previously reported for the gliosarcoma cases: 64%, 7/11 [6] and 20/28, 70% [11]. The frequency reported by us is similar to that of the previous findings (23%, 8/35) [14] and the frequency of *TP53* alterations in GBMs (28%) [19]. These differences may originate from a different methodology, the later study using whole exome sequencing on FFPE fixed tissues, a small cohort effect or different ratio of primary versus secondary gliosarcoma (in our cohort 9/10 gliosarcomas were primary).

CNV analysis showed a frequent amplification (40%, 4/10) of the chromosome seven region (Figure 2). Previous reports show a very low or absent *EGFR* amplification/overexpression in gliosarcoma tumors [6,14], but the recent copy-number analysis using CNV microarrays showed frequent *EGFR* amplification [11]. In our study, no focal amplification of the *EGFR* locus was found, just whole chromosome amplification.

We report the results of the transcriptomic analysis of gliosarcomas. The most striking difference between gliosarcomas and GBMs is the collagen gene signature, suggesting a more mesenchymal-like and extracellular matrix (ECM)-rich environment. Collagens type I (*COL1A1*, *COL1A2*), III (*COL3A1*) and VI (*COL6A2*, *COL6A3*) were highly upregulated in gliosarcomas, as seen in Figure 4C. The collagen-signature is reflected in the functional analysis of the groups of genes differentially expressed between GBMs and gliosarcomas, as “focal adhesion” is one of the discriminating groups (Table 3). COL6A3 appears to be a good marker of gliosarcoma tumors, as its expression was very high in the sarcomatous component, while it was virtually absent in the gliomatous component (Figure 6). Overexpression of genes related to integrin complexes ITGA5-ITGB1-CAL4A3 and ITGB1-NRP1 in gliosarcomas when compared to GBMs (Table 3) suggest that gliosarcomas are more migratory, invasive tumors, as these integrins are involved into epithelial to mesenchymal transition processes [34,35].

Interestingly, the top significant functional group of genes differentially expressed between GBMs and gliosarcomas belongs to the “Generic Transcription Pathway”, which groups mostly genes coding for zinc finger proteins (Figure 5). Most of these genes code for proteins that are involved in transcription and their expression is downregulated in gliosarcomas. Many zinc proteins are believed to be tumor suppressors [36]. Therefore, their down-regulation in gliosarcoma tumors could be interpreted as a signature of a more aggressive tumor phenotype. Many zinc proteins act as transcriptional repressors and their down-regulation may unlock the expression of genes that are commonly repressed in non-transformed cells [37,38].

An interesting translocation between *RABGEF1* and *GTF2IRD1P1* genes was discovered in three gliosarcoma samples. Due to the close proximity of these genes, it is most likely a long deletion or read-through transcript, as the distance between two fused RNA fragments is around three kB long. The last exon of the *RABGEF1* gene is merged with the *GTF2IRD1P1* gene, most likely causing its inactivation. As *RABGEF1* was linked to the development of some cancers [24,25] this alteration may have importance in gliosarcoma development. Further studies are needed to explain the impact of this translocation.

## 4. Materials and Methods

### 4.1. Tumor Samples

Ten fresh-frozen gliosarcoma samples and matching blood samples were obtained from the Departments of Neurosurgery, Institute of Psychiatry and Neurology (Warsaw, Poland), University Clinic Heidelberg University (Heidelberg, Germany) and Ludwig-Maximilians University (Munich, Germany). Each patient provided a written consent for the use of tumor tissues. Appendix A summarizes the clinical information about gliosarcoma patients. All procedures performed in studies involving human participants were in accordance with the ethical standards of the institutional and/or national research committee (#S-005/2003 by the Ethics Committee of Heidelberg University and #18-304 of the Ethics Committee of LMU University Clinics, Munich, Germany), with the 1964 Helsinki declaration and its later amendments.

### 4.2. DNA/RNA Isolation

Total DNA and RNA were extracted from fresh frozen gliosarcoma tissue samples using Trizol Reagent (Thermo Fischer Scientific, Waltham, MA, USA), following manufacturer’s protocol. Quality and quantity of nucleic acids were determined by Nanodrop (Thermo Fisher Scientific, Waltham, MA, USA) and Agilent Bioanalyzer (Agilent Biotechnologies, Santa Clara, CA, USA).

### 4.3. Design of Targeted Cancer-Related Gene Enrichment Panel

More than 90% of known pathogenic mutations registered in the NCBI ClinVar database are located in protein-coding DNA sequence (CDS) regions. We used SeqCap EZ Custom Enrichment Kit, which is an exome enrichment design that targets the latest genomic annotation GRCh38/hg38. Our SeqCap EZ Choice Library covers exomic regions of 664 genes frequently mutated in cancer. A majority of genes (578) were selected from the Roche Nimblegene Cancer Comprehensive Panel (based on Cancer Gene Consensus from Sanger Institute and NCBI Gene Tests). Additionally, we included 86 epigenetics-related genes (genes coding for histone acetylases/deacetylases, histone methylases/demethylases, DNA methylases/demethylases and chromatin remodeling proteins) based on a literature review [39,40,41].

### 4.4. DNA and RNA Sequencing

DNA isolated from tumor samples was processed for library preparation according to a NimbleGen SeqCap EZ Library SR (v 4.2) user guide. Detailed description in Appendix A: Total RNA library was prepared using SMARTER Stranded Total RNA Sample Prep Kit (Takara Clontech, Mountain View, CA, USA). Ribosomal RNA (rRNA) was removed from total RNA using RiboGone Technology (Takara Clontech, Mountain View, CA, USA). Briefly, one microgram of total RNAs were hybridized to rRNA-specific biotin labeled probes and digested by RNase H. The rRNA-free transcriptome RNA was concentrated by ethanol precipitation. The cDNA synthesis and DNA library construction for six gliosarcomas and eight GBMs were performed according to manufacturer’s protocol. Paired-end sequencing, resulting in 75 bases from each end of the fragments, was performed using Illumina HiSeq 1500 at the Nencki Institute core facility.

### 4.5. Data Analysis

#### 4.5.1. Sequence Alignment Pipeline

Fastq files from both tumor and blood samples were prepared as follows. Sequencing reads were filtered by trimmomatic program [42]. Filtered and trimmed reads were mapped to human genome version hg38 by BWA aligner (http://bio-bwa.sourceforge.net/, [43]). The mapped reads as the BAM files were sorted and indexed.

#### 4.5.2. Somatic Mutation Calling

Briefly, sorted and indexed BAM files from both tumor and blood DNA samples were used as an input for a Varscan2 analysis [44]. Somatic variants from Varscan2 for each tumor/blood sample pair were evaluated by Variant Effect Predictor (VEP) from Ensembl [45]. Variants that were assigned to the coding region of a gene and in the same time were described to be missense or nonsense variants, passed to further steps. In the final step, missense variants were evaluated using SIFT [46] and Polyphen2 [47] algorithms. The only variants that were described as non-tolerated/not-benign were selected to a final list of most likely pathogenic/damaging variants together with nonsense variants. The final list of the variants was stringently evaluated to comprise only the variants found in a region that was at least 30× covered in a tumor sample and a variant itself showed at least 20% penetration (20% or more reads presented the same mutation). There were some exceptions from this rule that will be discussed separately.

#### 4.5.3. Somatic Indel Calling

Based on BAM files as an input VarScan2 called indels and assigned them as Germline, Somatic and Unknown. Somatic indels were passed to further analysis and evaluated by Variant Effect Predictor (VEP) from Ensembl. Only indels assigned by VEP as coding indels were selected. The final list of indels was stringently evaluated to comprise of only the indels located within regions that were at least 20× covered in a tumor sample and a variant itself showed at least 15% penetration (15% or more reads were affected by the indel). Some exceptions from this rule will be discussed separately.

#### 4.5.4. Copy Number Variation (CNV) Calling

Sorted and indexed BAM files from both tumor and blood samples were used as an input for VarScan2 analysis in a CNV mode. Results from Varscan2 were segmented and prepared for visualization by DNA copy R library [48]. CNV profiles were visualized using GenVisR library (https://bioconductor.org/packages/release/bioc/html/GenVisR.html) [49].

#### 4.5.5. Tumor Purity

Tumor purity estimation (alpha value, assessing the proportion of tumor and normal tissue cells) was based on somatic mutations called by a VarScan2 program and CNV called by VarScan2 program in CNV mode. R library absCNseq [50] was used for the estimation of sample purity.

#### 4.5.6. RNA Expression and Functional Analyses

Sequencing reads (fastq) were filtered by the trimmomatic program, discarding all reads contaminated by the sequencing adapter sequence, as well as discarding reads in a bin of 20 bp reads had mean quality below Q30. The minimal length of the read used for mapping was set to 75 bp. Quality filtered reads were aligned to the human genome (hg38) by Tophat2 program [51]. Evaluation of transcriptomic profiles was performed using RSEM: RNA by Expectation and Maximization method [52]. Statistical analyses including comparisons to other samples (GBMs) were performed in an R environment.

Functional analysis was performed for the genes that were differentially expressed in GBM and gliosarcomass, as validated by the t-test (FDR corrected *p*-value below 0.05). The functional analysis of data was performed using g:Profiler to find statistically significant Gene Ontology terms, pathways and other gene function related terms [53]. Additionally, we have used the TCGA GBM gene expression signature (https://tcga-data.nci.nih.gov/docs/publications/gbm_exp/) and classified our samples to known subtypes.

#### 4.5.7. Translocations

Translocations in six gliosarcoma RNAseq samples were found using TopHat-Fusion software. Translocations called by the program were manually inspected in the IGV genome browser.

### 4.6. Immunohistochemical Staining

Most gliosarcoma samples (except one) were diagnosed as gliosarcomas in Heidelberg and Munich hospitals by combining reticulin and GFAP staining. One sample (GSM2) was diagnosed using a combined VIMENTIN and GFAP staining. For the detection of selected proteins, we performed a staining on 5-μm paraffin-embedded tissue sections. Sections were deparaffinized by incubation in xylene, ethanol (100, 90, 70%), and rehydrated. Epitopes were retrieved by oven boiling in a pH 6.0 citrate buffer for 40 min. Endogenous peroxidase was blocked in 0.3% H2O2 in methanol for 30 min followed by blocking with 10% horse serum. Sections were incubated overnight at 4 °C with mouse anti-COL6A3 (A-5) antibody (Santa Cruz Biotechnology, Dallas, TX, USA; dilution 1:200, 10% horse serum, 0.1% Triton X-100) or mouse anti-COL3A1 (B-10) antibody (Santa Cruz Biotechnology, dilution 1:200, 10% horse serum, 0.1% Triton X-100), respectively. Next, sections were washed in PBS, incubated with a biotinylated horse anti-mouse immunoglobulin (Vector, Labs, Burlingame, CA, USA; diluted 1:200 in 10% horse serum, 0.1% Triton X-100), then with horseradish peroxidase-conjugated avidin (ExtrAvidin™−Peroxidase, Sigma-Aldrich, Munich, Germany, dilution 1:200, PBS) for 60 min and with 3,3′-diaminobenzidine (DAB). Sections were stained with hematoxylin (Sigma-Aldrich, Munich, Germany), washed in PBS, and mounted. Immunohistochemical staining for GFAP was performed by an automatic stainer BenchMark Ultra (Ventana Medical Systems, Tucson, AZ, USA) using an anti-GFAP mouse primary antibody (Dako, dilution 1:100). To visualize the mesenchymal (sarcomatous) component of gliosarcoma, the Gomori’s silver staining was performed using a modified protocol [54].

More detailed methods are described in the Appendix A.

## 5. Conclusions

We demonstrate that most gliosarcoma tumors have somatic alterations of PIK3/Akt (*PTEN*, *PI3K*) and RAS/MAPK (*NF1*, *BRAF*) signaling pathways that are crucial for tumor growth and therapy resistance. Additionally, no gliosarcoma samples carried *IDH1/2* gene mutations, placing gliosarcomas among an IDH wild-type subtype of glioblastoma. Our findings demonstrate that gliosarcomas, in terms of somatic alterations, are fairly similar to GBMs, with a higher frequency of *NF1* and *PTEN* alterations, more similar to frequencies observed in mesenchymal GBMs. In terms of the transcriptomic profile, these tumors exhibit more of the mesenchymal signature, with different patterns of focal adhesion and cell invasion related genes. In the context of existing and emerging therapies, the current study shows that PI3K and BRAF inhibitors could be useful in targeted therapy for gliosarcoma patients, and that the molecular profiling of gliosarcoma could be very helpful to find fitting clinical trials or the off-label use of drugs.

## Figures and Tables

**Figure 1 cancers-11-00284-f001:**
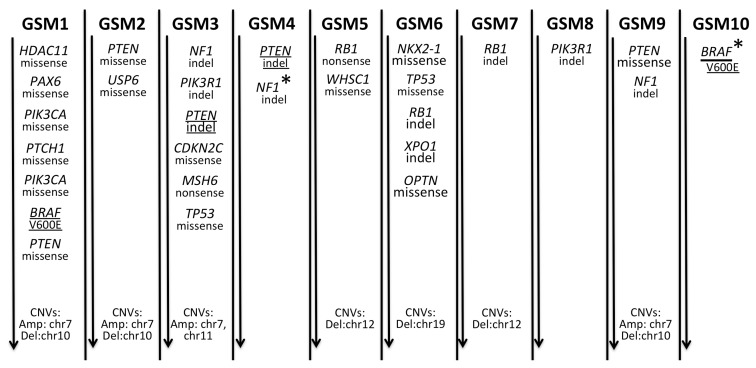
Summary of somatic genetic alterations in gliosarcomas. The panel represents somatically altered genes and chromosomal gain/loss in 10 gliosarcoma samples. Genetic alterations are ordered by decreasing the penetration in tumor tissue. Underlined alterations represent recurrent somatic alterations across analyzed samples, while an asterisk (*) marks alterations that were below 20% penetration threshold for somatic mutations or 15% penetration threshold for somatic indels. Those alterations were verified by ultra-deep sequencing.

**Figure 2 cancers-11-00284-f002:**
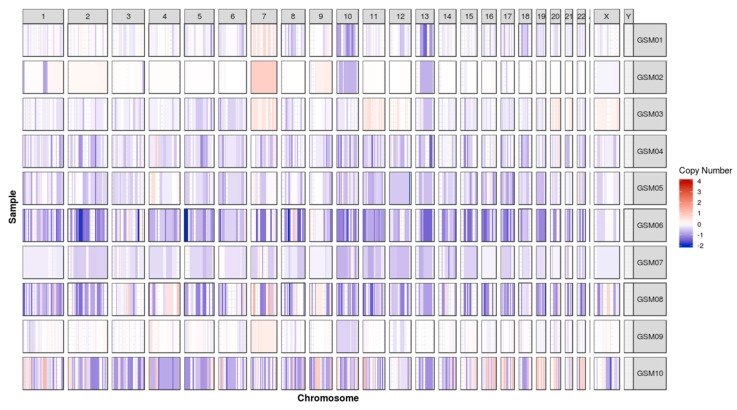
CNV plot for 10 gliosarcoma tumors. Scale is a log-ratio of a tumor versus paired normal (blood) sample across chromosomal locations (*X*-axis).

**Figure 3 cancers-11-00284-f003:**
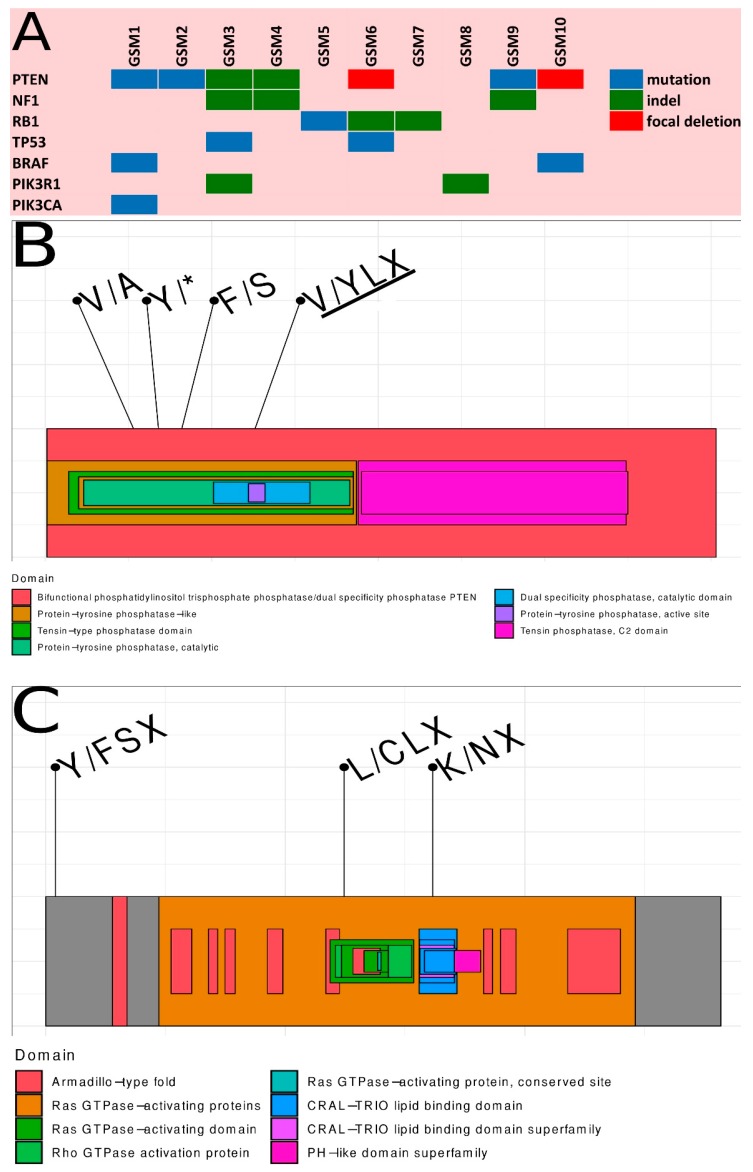
Detailed description of specific genetic alterations in each of the 10 samples. (**A**) Summary of somatic alterations with seven most frequently altered genes. (**B**) A view of the PTEN protein with marked positions of occurring somatic alterations. (**C**) A view of the NF1 protein with marked positions of occurring somatic alterations.

**Figure 4 cancers-11-00284-f004:**
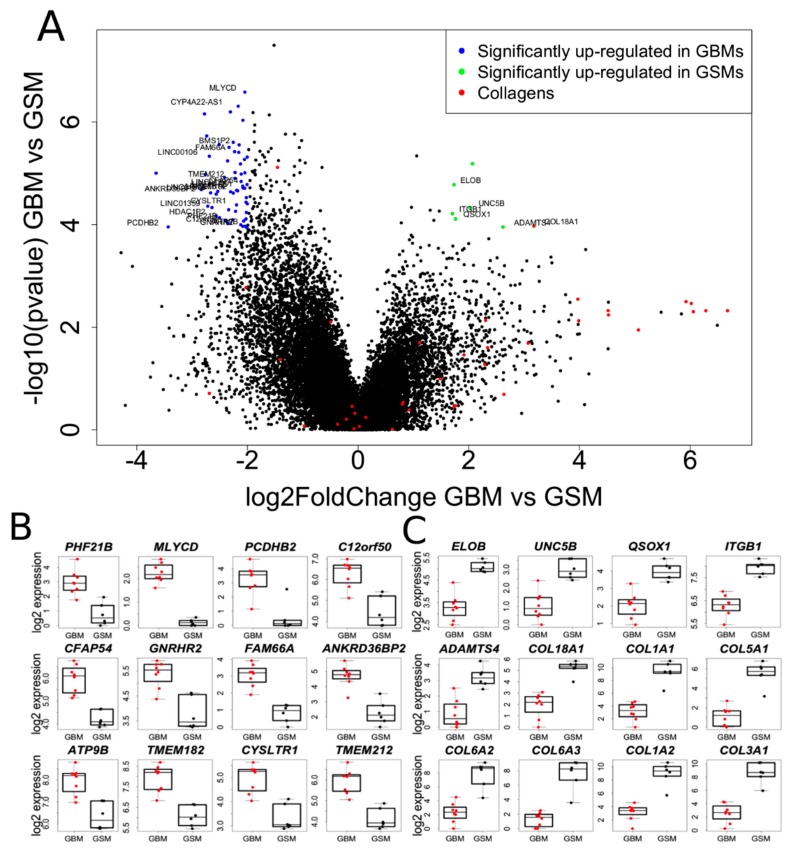
Characterization of gliosarcoma transcriptomes and comparison to glioblastoma (GBM). (**A**) Volcano plot of gene expression shows the differences between GBM and gliosarcoma. X axis depicts the significance of expression change (more distant from 0 being more significant) and Y axis depicts log2 fold change. Genes significantly up-regulated in GBM are color-coded in blue, while genes up-regulated in gliosarcoma are color-coded in green, collagen coding genes are color-coded in red. The threshold for color-coded genes where: log2FoldChange higher than 1.5 or lower than 0.5 and false discovery rate (FDR) adjusted *p*-value lower than 0.01. (**B**) Boxplot of selected genes that were down-regulated in gliosarcomas when compared to GBMs. (**C**) Boxplot of selected genes that were up-regulated in gliosarcomas when compared to GBMs with a special emphasis on collagens showing high fold difference between gliosarcomas and GBMs.

**Figure 5 cancers-11-00284-f005:**
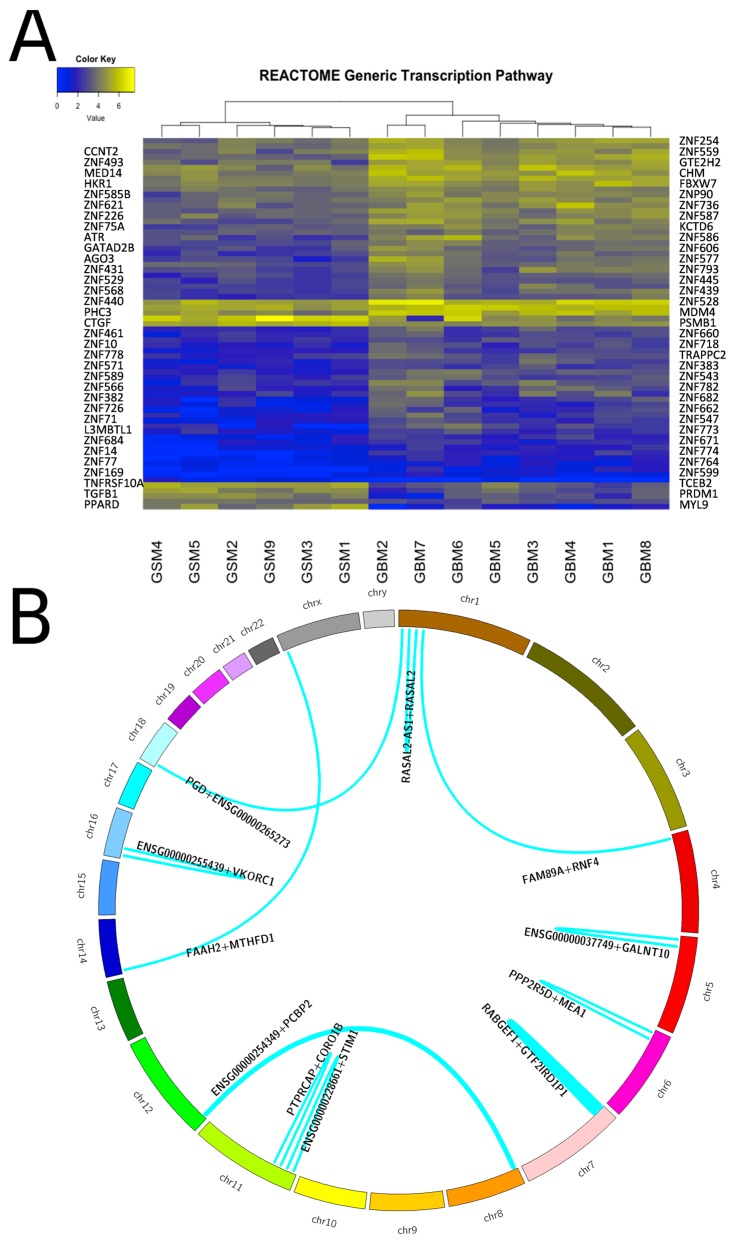
Transcriptomic analyses of gliosarcomas. (**A**). Heatmap of the Generic Transcription Pathway from REACTOME in GBM and gliosarcoma samples. Generic Transcription Pathway was the most significant functional group (defined by gProfiler) represented within genes differentially expressed between gliosarcoma and GBM samples. The heatmap was generated in R using heatmap.2 library with Euclidean distance measure; hierarchical clustering with complete agglomeration has been used. Gene names of odd numbered raws of heatmap is on the right side of the heatmap, while even numbered raws are on left side of the heatmap for better gene names visibility. (**B**) Circos plot of translocations detected in gliosarcoma samples. Line thickness is related to the number of samples detected with that translocation.

**Figure 6 cancers-11-00284-f006:**
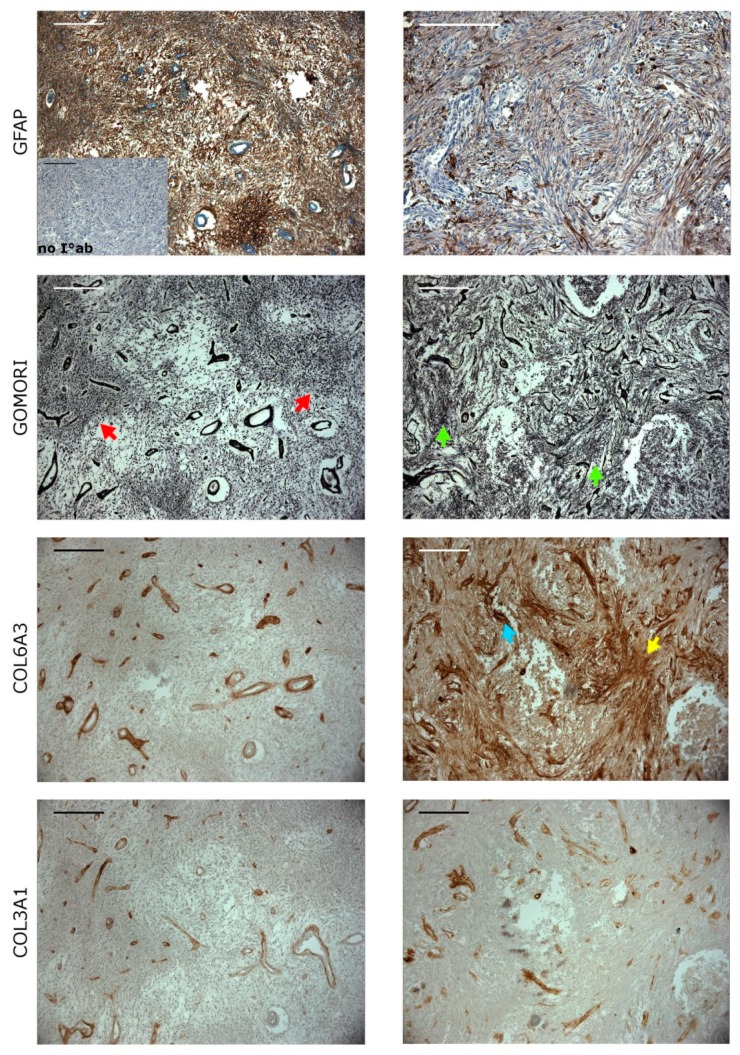
The expression of collagens in the gliomatous and sarcomatous components of gliosarcoma. The left panel shows GFAP positive staining (brownish area) and negative Gomori silver staining (red arrows) in the gliomatous tissue of gliosarcoma. COL6A3 and COL3A1 staining is restricted to the blood vessels of tissue. Positive Gomori (green arrows) and negative GFAP staining (bluish area), (the right panel), of spindle cells marks the sarcomatous component. COL6A3 is expressed mainly around spindle cells (yellow arrow) and blood vessels (blue arrow). COL3A1 is expressed mostly in blood vessels; Bars: 200 µm.

**Table 1 cancers-11-00284-t001:** Somatic mutations in gliosarcomas. Table is sorted by sample name and secondly by VAF (variant allele frequency) of mutation in a tumor sample. *p* value is a somatic status *p*-value from Varscan2 analysis. In bold—the only detected recurrent position of mutation in samples GSM1 and GSM10 in BRAF (V600E). Mutations in a table are all missense/nonsense mutations called as non-tolerated/non-benign by SIFT/Polyphen2 algorithms and are found with minimal penetration of a mutated allele >20% except one BRAF mutation marked as *. Selected variants have been additionally validated by Sanger sequencing, one variant failed to be validated by Sanger sequencing, most likely due to the fact, that validation had to be performed from different tissue specimen isolation. ND—it was not detected in primary Varscan2 analysis but it was found in secondary manual inspection.

Gene_Name	Chrom	Position	Reference	Variant	VAF	*p* Value	Sample	Sanger
*HDAC11*	chr3	13504480	C	T	39.13%	7.15 × 10^−7^	GSM1	validated
*PAX6*	chr11	31802832	G	A	38.66%	5.65 × 10^−4^	GSM1	not tested
*PIK3CA*	chr3	179198940	G	A	37.56%	8.01 × 10^−9^	GSM1	validated
*PTCH1*	chr9	95447237	C	T	36.81%	6.74 × 10^−4^	GSM1	not tested
***BRAF***	**chr7**	**140753336**	**A**	**T**	**26.68%**	**9.73 × 10^−5^**	**GSM1**	**validated**
*PTEN*	chr10	87894103	T	C	21.70%	6.02 × 10^−4^	GSM1	not validated
*PTEN*	chr10	87925552	C	A	55.32%	6.09 × 10^−6^	GSM2	validated
*USP6*	chr17	5170636	G	C	33.99%	1.19 × 10^−4^	GSM2	not tested
*CDKN2C*	chr1	50970430	T	G	60.38%	1.02 × 10^−14^	GSM3	validated
*TP53*	chr17	7673821	G	A	55.38%	1.53 × 10^−5^	GSM3	not tested
*MSH6*	chr2	47798725	C	T	53.85%	1.04 × 10^−12^	GSM3	validated
*RB1*	chr13	48342630	G	A	25.27%	1.65 × 10^−5^	GSM5	validated
*WHSC1*	chr4	1959574	C	T	22.05%	1.67 × 10^−3^	GSM5	not tested
*NKX2-1*	chr14	36519297	C	A	50%	1.25 × 10^−3^	GSM6	not tested
*TP53*	chr17	7674947	A	G	42.86%	2.45 × 10^−4^	GSM6	not tested
*OPTN*	chr10	13127868	G	T	27.27%	1.02 × 10^−2^	GSM6	not tested
*PTEN*	chr10	87960910	T	C	30.25%	2.38 × 10^−2^	GSM9	validated
***BRAF ****	**chr7**	**140753336**	**A**	**T**	**6.00%**	**ND**	**GSM10**	**not tested**

**Table 2 cancers-11-00284-t002:** Somatic indels in GSMs. Table is sorted by sample name and secondly by VAF (variant allele frequency) of indel in a tumor sample. *p* value is a somatic status *p*-value from Varscan2 analysis. In bold—the only detected recurrent position of the mutation in samples GSM3 and GSM4 in *PTEN*. All indels in a table are somatic coding indels that were found with minimal penetration of mutated allele >15% except one *NF1* indel marked as *. Selected variants were validated with Sanger sequencing.

Gene_Name	Chrom	Position	Ref	Variant	VAF	*p* Value	Sample	Sanger
*NF1*	chr17	31325895	G	-C	55.04%	6.9 × 10^−15^	GSM3	validated
*PIK3R1*	chr5	68280682	A	-CT	51.72%	8.2 × 10^−10^	GSM3	not tested
***PTEN***	**chr10**	**87961041**	**G**	**-TACT**	**50%**	**2.4 × 10^−11^**	**GSM3**	validated
***PTEN***	**chr10**	**87961041**	**G**	**-TACT**	**19.23%**	**1.1 × 10^−4^**	**GSM4**	validated
*NF1 **	chr17	31235638	C	-TGTT	14%	8.4 × 10^−3^	GSM4	validated
*RB1*	chr13	48456342	T	-A	34.88%	3.5 × 10^−5^	GSM6	not tested
*RB1*	chr13	48307352	C	-AG	17.28%	2.0 × 10^−2^	GSM7	not tested
*PIK3R1*	chr5	68293757	C	-ATGAAT	29.41%	1.6 × 10^−3^	GSM8	not tested
*NF1*	chr17	31159044	A	-TCTC	18.01%	1.6 × 10^−3^	GSM9	validated

**Table 3 cancers-11-00284-t003:** Functional gene groups significantly different between gliosarcoma ans glioblastoma samples. re—REACTOME, MF—GO molecular function, BP—GO biological process, CC—GO cellular component, co—CORUM, hp—Human Phenotype Ontology, ke—KEGG.

*p* Value	Term.id	Domain	Term.Name
2.69 × 10^−10^	REAC:212436	re	Generic Transcription Pathway
1.48 × 10^−7^	GO:0003676	MF	nucleic acid binding
4.31 × 10^−7^	GO:0046872	MF	metal ion binding
2.54 × 10^−6^	GO:0043169	MF	cation binding
0.000116	GO:0003677	MF	DNA binding
0.000167	REAC:74160	re	Gene Expression
0.000202	GO:0044260	BP	cellular macromolecule metabolic process
0.000492	GO:0090304	BP	nucleic acid metabolic process
0.000769	GO:0016070	BP	RNA metabolic process
0.00198	GO:0043170	BP	macromolecule metabolic process
0.00417	GO:0005925	CC	focal adhesion
0.0058	GO:0005924	CC	cell-substrate adherens junction
0.00701	GO:0006139	BP	nucleobase-containing compound metabolic process
0.00751	GO:0030055	CC	cell-substrate junction
0.00796	GO:0005634	CC	nucleus
0.0227	GO:0043227	CC	membrane-bounded organelle
0.0234	CORUM:2853	co	ITGA5-ITGB1-CAL4A3 complex
0.0267	GO:1901363	MF	heterocyclic compound binding
0.0339	GO:0009059	BP	macromolecule biosynthetic process
0.0346	GO:0010467	BP	gene expression
0.0366	GO:0043231	CC	intracellular membrane-bounded organelle
0.0391	GO:0051252	BP	regulation of RNA metabolic process
0.0463	CORUM:3104	co	ITGB1-NRP1 complex
0.05	CORUM:5658	co	Nrp1-PlexinD1 complex
0.05	HP:0002693	hp	Abnormality of the skull base
0.05	KEGG:00511	ke	Other glycan degradation

**Table 4 cancers-11-00284-t004:** Translocations in GSM samples. In bold—translocations discovered in 2 samples, in bold with *—translocations discovered in 3 samples.

Sample	gene1	Chr_gene1	Position_gene1	gene2	Chr_gene2	Position_gene2
GSM1	*PTPRCAP*	chr11	67204279	*CORO1B*	chr11	67206320
**GSM1**	***RABGEF1 ****	**chr7**	**66273872**	***GTF2IRD1P1***	**chr7**	**66275833**
GSM2	*MTHFD1*	chr14	64909104	*FAAH2*	chrX	57419899
GSM3	*ENSG00000037749*	chr5	153569748	*GALNT10*	chr5	153674375
GSM3	*ENSG00000228661*	chr11	3876644	*STIM1*	chr11	3988780
GSM3	*FAM89A*	chr1	231157568	*RNF4*	chr4	2514808
**GSM3**	***RABGEF1 ****	**chr7**	**66273872**	***GTF2IRD1P1***	**chr7**	**66275833**
**GSM5**	***ENSG00000254349***	**chr8**	**75515899**	***PCBP2***	**chr12**	**53858635**
GSM5	*PGD*	chr1	10464318	*ENSG00000265273*	chr18	29542460
GSM5	*RASAL2-AS1*	chr1	178063003	*RASAL2*	chr1	178063501
**GSM9**	***ENSG00000254349***	**chr8**	**75515899**	***PCBP2***	**chr12**	**53858635**
GSM9	*ENSG00000255439*	chr16	31102095	*VKORC1*	chr16	31102662
**GSM9**	***RABGEF1 ****	**chr7**	**66273872**	***GTF2IRD1P1***	**chr7**	**66275833**

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
