# Peer review of "Gliosarcoma Is Driven by Alterations in PI3K/Akt, RAS/MAPK Pathways and Characterized by Collagen Gene Expression Signature"

_cancers, 2019, doi:10.3390/cancers11030284_

Round 1

Reviewer 1 Report

In this manuscript, the authors investigated on gliosarcoma indentifying a profiling of genes differentially expressed vs glioblastoma. Despite Gliosarcoma is a rare type of cancer is lethal and characterizing this cancer is really important also because is largelly unknown both on clinical and biological profile. I want to congratulate with the authors for the work done

Author Response

Thank you very much for your kind words!

Reviewer 2 Report

General comment:

I think that this issue is very interesting as well as the authors’ findings. There are a lot of work and different techniques done in this study.

However, the manuscript needs some clarifications/corrections.

·         The manuscript is not according with new CNS 2016 classification, in 2016 Gliosarcomas was classified with GBM-IDH-wildtype variant. I think it should the altered in text. In the title in not necessary to referred IDH-wildtype GBM because it’s a redundancy. For exemple, in line 53 it is necessary to update WHO 2016.

·         Line 77-79: I think there is a mistake Oh JE et al 2016 did not used low grade gliomas to gliosarcoma, they used primary gliosarcoma.

·         Regarding Results: It is not clear the reason by which only some variants were validated and confirmed by sanger seq, and in table 1 did not appear all mutations found, for instance, in PIK3CA.

·         Figures are deformed and is difficult to follow, in fig 1 I can not visualize * and appears question markers(?).

·         Line 129 what does it mean 3/10? 3 samples of GSM? But in table 2 only appears sample GSM3

·         All tables present asterisks and it is confusing to follow, I suggest adding new column with the number of samples with the present mutation/alteration.

·         Figure 3 is deformed

·         Over manuscript appears Figure xA and yb. (ex: 172 -173 vs 192)

·         Figure 5 needs improvement in quality, I cannot read the genes

·         Regarding transcriptomic analysis I think that the presentation of the results could be improved: first to show pathways and at the end to show specific genes, which should be more investigated. So, I think it could be easier to follow presenting figure 5, table 3 and then figure 4 its confirmation by IHC by figure 6. Figure 4 and 6 could be together. However, I think is missing IHC in GBM cases to prove that collagens expressions are specific of GSM.

·         In figure 6 is missing arrows to better follow the legend and text.

·         Could you use databases to achieve the biomarkers value of collagens in GSM, as a specific biomarker of GSM vs GBM? To improve your results.

·         Discussion: It is extensively discussed the similarities between GSM and GBM and less the differences. The most interesting results regarding collagens and new translocation found are less discussed. I think that authors should alter: refer to GBM IDH wt. When you compare the findings in this study with other studies pay attention if the studies had referred is about GBM total or IDH wt or mut, I think it is important. To refer the mutations found in PIK3CA gene and mention the first study that described mutations in PIK3CA in gliomas (Samuel et al 2004). It is missing to discuss the putative target inhibition of some targets found.

Author Response

General comment: I think that this issue is very interesting as well as the authors’ findings. There are a lot of work and different techniques done in this study. However, the manuscript needs some clarifications/corrections.

·        The manuscript is not according with new CNS 2016 classification, in 2016 Gliosarcomas was classified with GBM-IDH-wildtype variant. I think it should the altered in text. In the title in not necessary to referred IDH-wildtype GBM because it’s a redundancy. For example, in line 53 it is necessary to update WHO 2016.

We thank the referee for this remark. It has been corrected in the revised text. We do agree that it is not necessary in the title, s it has been changed.

·        Line 77-79: I think there is a mistake Oh JE et al 2016 did not used low grade gliomas to gliosarcoma, they used primary gliosarcoma.

We thank the referee for pointing this mistake. It is corrected in the revised version of the manuscript.

·        Regarding Results: It is not clear the reason by which only some variants were validated and confirmed by sanger seq, and in table 1 did not appear all mutations found, for instance, in PIK3CA.

As Sanger sequencing, is still considered a gold standard in clinical settings, we validated several variants using that metholdology . We were unable to test all of them due to a low amount of material in some samples, as most of the material were used for targeted NGS. We considered findings of genetic alterations in PTENand NF1genes the most interesting (due to different frequencies than in GBM) and validated those findings. We performed validation of some additional variants (mutBRAF) were was possible based on the existing clinical material.

We propose adding following sentence at the end of section 2.1 of Results:

Sanger sequencing was performed to confirm PTENand NF1genetic alterations as they were the most frequent ones. Additional gene variants were validated based on availability of the clinical material.

·        Figures are deformed and is difficult to follow, in fig 1 I can not visualize * and appears question markers(?).

We apologize for insufficient quality of figures due to size reduction. To increase its quality and visibility, we  increased an asterix font size in the revised version of the manuscript. There is no question mark in the figure 1.

·        Line 129 what does it mean 3/10? 3 samples of GSM? But in table 2 only appears sample GSM3

It means 3 out of 10 samples. To make is clear we changed it to 3/10 gliosarcoma samples. NF1 mutation was found in 3 samples: GSM3, GSM4 and GSM9 as in the Table 2.

·        All tables present asterisks and it is confusing to follow, I suggest adding new column with the number of samples with the present mutation/alteration.

We have followed referee’s suggestion and added one additional column with information about Sanger validation and changed Table descriptions accordingly.

·        Figure 3 is deformed

We provided the new fig.3 with hopefully better organization of panels and more visible details This deformation arose from transferring documents encoded in a MAC OS system versus Windows system, which deformed this graph. If possible, it will be uploaded separately.

·        Over manuscript appears Figure xA and yb. (ex: 172 -173 vs 192)

It has been corrected in the revised version.

·        Figure 5 needs improvement in quality, I cannot read the genes

We have improved the quality and gene names size. Heatmap was described with gene names from both sides to increase font size, so hopefully the gene names are visible now. The following sentence was added in the figure legend:

Gene names of odd numbered raws of heatmap is on the right side of heatmap, while even numbered raws are on left side of heatmap for better gene names visibility.

·        Regarding transcriptomic analysis I think that the presentation of the results could be improved: first to show pathways and at the end to show specific genes, which should be more investigated. So, I think it could be easier to follow presenting figure 5, table 3 and then figure 4 its confirmation by IHC by figure 6. Figure 4 and 6 could be together. However, I think is missing IHC in GBM cases to prove that collagens expressions are specific of GSM.

We thank for this useful suggestion. We have tried to re-arrange the figure panels and such changes didi not improve the picture. Searching for translocations and functional analysis are extensions of RNAseq analysis, therefore we believe that the provided presentation is better than one the proposed by the referee. We did not merge Figure 4 and 6 because confirmation at the protein level is a separate issue and IHC images  would be too small to appreciate differences. In this case it was important to performed control staining for different compartments of gliosarcoma. Images of staining of collagens in GBM cases are provided  in the Appendix A, Figure 6A.

·        In figure 6 is missing arrows to better follow the legend and text.

Thank you for the suggestion, arrows have been added.

·        Could you use databases to achieve the biomarkers value of collagens in GSM, as a specific biomarker of GSM vs GBM? To improve your results.

This is an interesting suggestion, but we are not aware of any database that could help us in that matter. Gliosarcoma was not included in the TCGA database and there is no other transcriptomic data on this tumor. We can try to compare our data with TCGA data on GBM to see if collagens are differently expressed, but due to different platforms, different technology and normalization, direct comparison with our data would be difficult and not fully reliable. We did however find COL3A1 to be not detected in any of staining from protein atlas, we have added this information as Appendix_D.

Following sentence was added in the Results section:

COL3A1 is most likely a very specific marker of gliosarcoma compartment as it was found not detected in any of tissues in immunochemistry results from Protein Atlas (Appendix D).

·        Discussion: It is extensively discussed the similarities between GSM and GBM and less the differences. The most interesting results regarding collagens and new translocation found are less discussed. I think that authors should alter: refer to GBM IDH wt. When you compare the findings in this study with other studies pay attention if the studies had referred is about GBM total or IDH wt or mut, I think it is important. To refer the mutations found in PIK3CA gene and mention the first study that described mutations in PIK3CA in gliomas (Samuel et al 2004). It is missing to discuss the putative target inhibition of some targets found.

Aa a referee pointed out the most interesting results presented here refer to transcriptomic analyses and different expression of collagens in GSM and GBM. We hint that all presented differences make gliosarcoma more than a variant of GBM but we were afraid to put a more broad statement. We believe that new translocation found are sufficiently discussed, as at the moment we have not more data to speculate on their specific roles.  

We agree that the difference between IDH-wildtype and IDH-mutant group is important, but in many previous studies it was not underlined or not discussed which type of GBM is described. Primary and secondary glioblastomas were frequently mixed in one group of samples and analyzed as one group. After 2016 WHO classification, these groups are more strictly separated. It has been revised in the current version of manuscript.

Thank you for information regarding Samuel et al. publication, it has been added to the reference list and cited in discussion.

We have underlined the presence of the BRAFgene mutation, which it is druggable by mutated BRAF inhibitors. Sadly, targeting PTENaltered gliomas is not yet achieved, apart from some preliminary preclinical studies, therefore to our best knowledge, there is no other drugs in the clinics that could be used.